# Early Visibility of Cellular Aggregates and Changes in Central Corneal Thickness as Predictors of Successful Corneal Endothelial Cell Injection Therapy

**DOI:** 10.3390/cells12081167

**Published:** 2023-04-15

**Authors:** Evan N. Wong, Valencia H. X. Foo, Gary S. L. Peh, Hla M. Htoon, Heng-Pei Ang, Belinda Y. L. Tan, Hon-Shing Ong, Jodhbir S. Mehta

**Affiliations:** 1Corneal and External Diseases Department, Singapore National Eye Centre, Singapore 169856, Singapore; 2Tissue Engineering and Cell Therapy Group, Singapore Eye Research Institute, Singapore 169856, Singapore; 3Centre for Ophthalmology and Visual Science (Incorporating Lions Eye Institute), The University of Western Australia, Perth, WA 6009, Australia; 4Department of Ophthalmology and Visual Science, Duke-National University of Singapore (NUS) Graduate Medical School, Singapore 169857, Singapore; 5School of Material Science & Engineering, Nanyang Technological University, Singapore 639798, Singapore

**Keywords:** ophthalmology, cornea, corneal endothelium, bullous keratopathy, cell injection, cell therapy, anterior-segment optical coherence tomography, regenerative medicine, corneal transplantation, tissue engineering

## Abstract

(1) Background: Cell injection therapy is an emerging treatment for bullous keratopathy (BK). Anterior segment optical coherence tomography (AS-OCT) imaging allows the high-resolution assessment of the anterior chamber. Our study aimed to investigate the predictive value of the visibility of cellular aggregates for corneal deturgescence in an animal model of bullous keratopathy. (2) Methods: Cell injections of corneal endothelial cells were performed in 45 eyes in a rabbit model of BK. AS-OCT imaging and central corneal thickness (CCT) measurement were performed at baseline and on day 1, day 4, day 7 and day 14 following cell injection. A logistic regression was modelled to predict successful corneal deturgescence and its failure with cell aggregate visibility and CCT. Receiver-operating characteristic (ROC) curves were plotted, and areas under the curve (AUC) calculated for each time point in these models. (3) Results: Cellular aggregates were identified on days 1, 4, 7 and 14 in 86.7%, 39.5%, 20.0% and 4.4% of eyes, respectively. The positive predictive value of cellular aggregate visibility for successful corneal deturgescence was 71.8%, 64.7%, 66.7% and 100.0% at each time point, respectively. Using logistic regression modelling, the visibility of cellular aggregates on day 1 appeared to increase the likelihood of successful corneal deturgescence, but this did not reach statistical significance. An increase in pachymetry, however, resulted in a small but statistically significant decreased likelihood of success, with an odds ratio of 0.996 for days 1 (95% CI 0.993–1.000), 2 (95% CI 0.993–0.999) and 14 (95% CI 0.994–0.998) and an odds ratio of 0.994 (95% CI 0.991–0.998) for day 7. The ROC curves were plotted, and the AUC values were 0.72 (95% CI 0.55–0.89), 0.80 (95% CI 0. 62–0.98), 0.86 (95% CI 0.71–1.00) and 0.90 (95% CI 0.80–0.99) for days 1, 4, 7 and 14, respectively. (4) Conclusions: Logistic regression modelling of cell aggregate visibility and CCT was predictive of successful corneal endothelial cell injection therapy.

## 1. Introduction

Corneal transparency is a prerequisite for good vision. Corneal endothelial cells (CEC) play a crucial role in maintaining the cornea in a state of normal transparency [1]. In the healthy cornea, CECs form a monolayer on the posterior surface of the cornea. They serve two main functions. Firstly, they provide a pathway for nutrient uptake and waste product removal. This is facilitated by simple diffusion and secondary active transport. Secondly, they serve as a fluid pump. The active transport of ions into the aqueous humour draws water out by osmotic forces [2]. This second function counteracts a continuous leak of fluid into the stroma, which is generated by imbibition pressure from stromal hydrophilic glycosaminoglycans. In this way, corneal transparency is maintained in a state of relative deturgescence [2]. Excessive CEC loss may arise from accidental or surgical traumas, corneal dystrophies or infections. This may result in loss of barrier function, fluid accumulation, reduction of transparency and the formation of epithelial bullae [3,4,5]. The most common causes for endothelial dysfunction are Fuch’s endothelial corneal dystrophy (FECD) and pseudophakic bullous keratopathy (PBK) [6,7,8].

Endothelial keratoplasty has superseded penetrating keratoplasty as the current gold standard of treatment for the loss of corneal clarity secondary to endothelial failure [9]. Various evolving techniques have been developed, beginning in the late 1990s with Deep Lamellar Endothelial Keratoplasty (DLEK), followed by Descemet Stripping Endothelial Keratoplasty (DSEK) in 2003, Descemet Stripping Automated Endothelial Keratoplasty (DSAEK), which involved the preparation of the donor graft with a microkeratome, ultra-thin DSAEK (UT-DSAEK) and, currently, Descemet Membrane Endothelial Keratoplasty (DMEK) [10,11]. However, there is a global shortage of transplant-grade corneal tissue [12]. A survey of corneal transplants performed in 116 countries between 2012 and 2013 estimated that there was only one donor cornea for every 70 required [6]. The worldwide trends of increasing life expectancy and ageing populations may only serve to widen the supply–demand mismatch, as the absolute number of older individuals requiring transplants increases, while the number of available healthy younger donors decreases [8,13]. There is a significant impetus, therefore, to develop alternative and non-donor-reliant solutions to treating endothelial failure.

Cell-based therapies represent a potential disruptive technology for this clinical indication. CECs are post-mitotic cells arrested in the G1 phase of the cell cycle [5]. They are typically maintained in a non-proliferative state in vivo but have been shown to proliferate in vitro [4,14]. Numerous groups have reported a variety of cell sources and methods to achieve functional CECs, including deriving CECs from the expansion of primary donor cadaveric CECs [15,16,17,18], human embryonic stem cells (hESC) [19,20,21,22] and human induced pluripotent stem cells (hiPSC) [23,24,25]. Ong et al. also recently reported functional outcomes with an alternative method that did not require complex cellular propagation techniques. This approach, termed Simple Non-cultivated Endothelial Cells (SNEC), involved the isolation of donor DM/CECs, their stabilization for 48 h and the dissociation of CECs into single cells with mild enzymatic digestion. These cells could then be pooled to a sufficient concentration in preparation for the treatment [26]. Whether cultivated or non-cultivated, these cells can then be delivered into the anterior chamber of the recipient’s eye, either by seeding them onto a tissue-engineered graft or by direct injection [27]. Pre-clinical studies have reported safety and efficacy outcomes in primate and rabbit models [16,28]. Subsequently, Kinoshita et al. reported successful outcomes in a small clinical series, with a 5-year follow-up [29,30]. 

Anterior segment optical coherence tomography (AS-OCT) is an imaging modality that allows for high-resolution cross-sectional and semi-quantitative imaging of the cornea, anterior chamber and iridocorneal angle. It has been shown to be useful in imaging the layers of the cornea after anterior and posterior lamellar keratoplasty [31]. Various AS-OCT parameters have been investigated for usefulness in monitoring DMEK and DSAEK grafts, such as pachymetry parameters, anterior chamber depth and volume, angle parameters and corneal densitometry parameters [32,33,34]. It was noted on routine post-CEC injection (CECI) AS-OCT imaging of rabbit corneas that small, hyperreflective foci were present in some, but not all, eyes. These were presumed to be consistent with injected single cells subsequently forming aggregated cellular clusters that were visible on AS-OCT imaging.

We hypothesized that the early detection by AS-OCT imaging of cellular aggregates on Descemet’s membrane following CECI could predict the eventual corneal deturgescence in a rabbit model of bullous keratopathy. Such imaging-based prediction could guide early interventions, such as re-injection. 

## 2. Materials and Methods

### 2.1. Study Design

This retrospective study was part of a larger project approved by the local centralized institutional review board (Singhealth IRB Ref: 2013/783/A; and 2016/2839). The objective of this study was to determine whether the detection of cellular aggregates by AS-OCT imaging was predictive of the eventual functional success of CECI therapy. 

### 2.2. Materials

Ham’s F12, Medium 199, Human Endothelial-SFM, Dulbecco’s Phosphate-Buffered Saline (PBS), TrypLETM Select (TS), gentamicin, amphotericin B, penicillin and streptomycin were purchased from Life Technologies (Carlsbad, CA, USA). Insulin/Transferrin/Selenium (ITS) was purchased from Corning (New York, NY, USA), and ascorbic acid from Avantor (Pennsylvania USA). Collagen IV from human placenta and Trypan blue (0.4%) were purchased from Sigma (St. Louis, MO, USA). Recombinant human basic fibroblast growth factor and rho-associated, coiled-coil protein kinase inhibitor Y-27632 were brought from Miltenyi Biotec (Bergisch Gladbach, Germany). The FNC coating mixture was obtained from United States Biologicals (Swampscott, MA, USA). Liberase TH was purchased from Roche (Mannhein, Germany). EquaFetal^®^ was from Atlas Biologicals (Fort Collins, CO, USA). 

### 2.3. Research-Grade Human Corneo-Scleral Tissues

All research-grade human cadaver corneal tissues were procured from either the Lions Eye Institute for Transplant and Research (Tampa, FL, USA) or Miracles in Sights (Winston-Salem, NC, USA), with informed consent from the next of kin. All research performed with human-derived tissue was carried out in accordance with the principles outlined in the Declaration of Helsinki. All corneo-scleral donor tissues were preserved and transported in Optisol-GS (Bausch & Lomb, New York, NY, USA) at 4 °C until they were processed.

### 2.4. Primary Corneal Endothelial Cells for CECI

#### 2.4.1. Expanded Primary CECs for CECI

Propagation of CECs for this study was achieved using a dual media culture approach as described [15,35]. Briefly, CECs were isolated using a two-step enzymatic digestion to first dislodge the CECs from the DM, followed by a second brief 5 min incubation step to further dissociate the cellular clusters into smaller clumps. The isolated cells were briefly rinsed twice before being seeded onto pre-coated collagen culture vessels and established in a corneal endothelial maintenance/stabilization medium (M5-Endo; Human Endothelial-SFM supplemented with 5% serum) overnight. Subsequently, CECs were cultured in the proliferative medium (M4-F99; Ham’s F12/M199, 5% serum, 20 μg/mL of ascorbic acid, 1x ITS, and 10 ng/mL of HrFGF) to promote the proliferation of the attached CECs. Once CECs reached 80% to 90% confluence, they were re-exposed to M5-Endo for at least two days before being dissociated using TS into a single-cell suspension. From here, the dissociated CECs were either re-plated at a seeding density of at least 1.0 × 10^4^ cells per cm^2^ on pre-coated collagen surfaces for further expansion or prepared in a final concentration of 500,000 cells within a volume of 120 µL of M5-Endo containing a Rho-associated kinase inhibitor (ROCKi) for CECI. Primary CECs were expanded to the second passage for the cell injection studies. All cultures were incubated in a humidified atmosphere at 37 °C with 5% CO_2_.

#### 2.4.2. Primary ‘Simple Non-Expanded Endothelial Cells’ (SNECs) for CECI

For the SNEC-harvesting approach, the Descemet membrane/corneal endothelium (DM/CE) from paired donor corneas was carefully peeled under a stereoscopic dissecting microscope as previously described [15]. Briefly, isolated DM/CE pieces were pooled and incubated in M5-Endo medium for at least 48 h. Following incubation (after 48 h), the M5-Endo medium was replaced with TS, and incubation continued for between 30 to 50 min, with a periodic check of the status of trypsinization (based on the cellular morphology of CECs), every 10 min after the first 20 min, as the rate of trypsinization varies depending on the donor. Upon rounding up of the majority of the CECs, the DM/CE pieces were lightly agitated with a 1 mL pipette before a final incubation of 5 min. Thereafter, the trypsinized single cells were collected and passed through a pre-wet 100 µm filter, before being prepared in a final volume of 120 µL of M5-Endo containing ROCKi.

### 2.5. Animal Surgeries

New Zealand White rabbits (n = 45) were used for this study. Lens extraction surgeries and cell injection procedures were performed by JSM, HSO and EW. The use of the rabbits, together with their care and treatment, strictly adhered to the regulations of the ARVO statement for the Use of Animals in Ophthalmic and Vision Research. All experimental procedures were approved by the Institutional Animal Care and Use Committee of SingHealth, Singapore (Ref: 2017/SHS/1343). All surgical procedures and follow-up evaluations were performed under general anesthesia achieved by intramuscular injections of 5 mg/kg of xylazine hydrochloride (Troy Laboratories, New South Wales, Australia) and 50 mg/kg of ketamine hydrochloride (Parnell Laboratories, New South Wales, Australia), along with topical application of lignocaine hydrochloride 1% (Pfizer Laboratories, New York, NY, USA). The crystalline lenses of the rabbits were extracted using a standard phacoemulsification technique [35]. To achieve mydriasis, tropicamide 1% (Alcon Laboratories, TX, USA) and phenylephrine hydrochloride 2.5% (Alcon Laboratories) eye drops were administered approximately 30 minutes before surgery. A clear corneal incision was made with a 2.8 mm disposable keratome. A 5.0 mm-diameter continuous curvilinear capsulotomy of the anterior capsule was created under ophthalmic viscosurgical devices (Viscoat; Alcon Laboratories) instilled into the AC. Hydro-dissection was performed using a 27-gauge cannula. The lens was then aspirated and removed with a standard phacoemulsification procedure using the White Star phacoemulsification system (Abbott Medical Optics, CA, USA). Subsequently, the corneal incision was sutured with a 10/0 nylon suture, and the rabbits were left aphakic with an intact posterior capsule for at least one week before the cell injection procedures. All rabbits were followed for 21 days after surgery before being sacrificed under anesthesia with an overdose of intracardiac injection of 85 mg/kg of sodium pentobarbitone (Jurox, New South Wales, Australia).

### 2.6. Corneal Endothelial Cell Injection

The method for CECI is based on our previous studies on the functional evaluation of cell-based therapies [36]. Briefly, prior to cell injection, a single intravenous dose of heparin (500 units in 1.0 mL) was administered to the rabbits to reduce intraocular fibrin formation. Subsequently, an AC maintainer was placed to infuse a balanced salt solution (BSS) containing additional heparin (1 unit per ml). A paracentesis was then created with a diamond knife to accommodate the insertion of a 30-gauge silicone soft-tipped cannula (catalogue number: SP-125053, ASICO, IL, USA) for the scraping of the rabbits’ CECs. The aim was the full removal of all rabbits’ CECs from limbus to limbus, whilst keeping the DM intact. A solution of trypan blue was then injected intracamerally to aid in the assessment of the DM denudation, and any residual CEs were re-scraped. Subsequently, 0.5 mL of 100 μg/mL carbochol (Miostat^®^, Alcon Laboartories) was injected to achieve intraoperative miosis. Both the paracentesis incision and the AC maintainer paracentesis sites were secured with 10/0 nylon interrupted sutures. This was followed by a 0.2 mL anti-inflammatory and anti-infective subconjunctival injection of a 1:1 mixture of 4 mg/mL of dexamethasone sodium phosphate (Hospira, Melbourne, VI, Australia) and 40 mg/mL of gentamicin sulfate (Shin Poong Pharmaceutical, Seoul, Korea). Using a syringe and 30 G cannula, 0.4 mL of aqueous humour was removed to shallow the anterior chamber. The harvested CECs suspended in ROCKi (Y-27632, AR-13324 or AR-13503) and M5-endo medium were then injected through a separate tunneled track via a 30 G needle. Immediately following cell injection, the rabbits were placed in a manner that ensured the cornea was in a downward position and maintained for three hours under volatile anesthesia.

### 2.7. Post-Operative Care

Following the cell injection procedures, all rabbits received a post-operative regime of topical prednisolone acetate 1% (Allergan Inc, NJ, USA) and topical antibiotic tobramycin 1% (Alcon Laboratories) four times a day. An intramuscular injection of 1 mL/kg of dexamethasone sodium phosphate (Norbrook Laboratories, Northern Ireland, UK) was also administered once daily. This medication regime was maintained until the rabbits were sacrificed.

### 2.8. Corneal Imaging and Intra-Ocular Pressure Measurement

All corneal imaging and measurements of intra-ocular pressure (IOP) were performed at baseline prior to cell injection, as well as on days 1 and 4 and in weeks 2 and 3 after cell injection. Corneal cross-sectional scans were performed using AS-OCT (Optovue, Fremont, CA, USA). The measurement of the central corneal thickness (CCT) was performed using calipers on the in-built Optovue software. Slit lamp photographs were taken with a Zoom Slit Lamp NS-2D (Righton, Tokyo, Japan). 

### 2.9. Statistical Analysis

The data were analyzed using Statistical Program for Social Sciences (SPSS©) Version 22 (IBM, New York, NY, USA). Differences in the distribution of continuous variables between groups were analyzed using the two-tailed independent *t*-test. The significance level was set at *p* < 0.05. A logistic regression was modelled to predict successful corneal deturgescence and its failure with cell visibility of aggregates and CCT. Receiver-operating characteristic (ROC) curves were plotted, and the areas under the curve (AUC) calculated for each time point from these models. A sensitivity analysis was performed to examine differences in cell visibility and pachymetry between eyes that received expanded primary CEC or SNECi for CECI using Pearson’s chi-square and Mann–Whitney U tests.

## 3. Results

### 3.1. Pre-Operative Assessment of Rabbits Following Lens Extraction

Forty-five right eyes were involved in this study. Thirty-seven eyes received CECI with expanded primary CEC and eight eyes received CECI with SNECi CECs. All rabbits were assessed prior to CECI, and at least one week following lens extraction surgery. Prior to the procedure, the eyes of all the rabbits were uninflamed, and the corneas were clear, with no visible corneal opacities or vascularization. No intraocular inflammation was observed. The mean (SD) pre-CECI CCT of the corneas was 367.1 (41.4) μm. 

### 3.2. Post-Operative Assessment of Clinical Outcomes in a Rabbit Model of Bullous Keratopathy

Successful corneal deturgescence was defined as a reduction in CCT at any time point within the first two weeks, compared to CCT on day 1. The percentage of eyes with successful deturgescence was 68.9% (31/45 eyes). For eyes with successful deturgescence, the mean (SD) CCT was 714.3 (180.7), 653.7 (217.5), 566.7 (172.2) and 637.2 (275.1) μm for days 1, 4, 7 and 14, respectively. For eyes with failed deturgescence, the mean (SD) CCT was 847.1 (245.7), 1100.2 (438.8), 1086.8 (455.7) and 1282.2 (478.2) μm for days 1, 4, 7 and 14, respectively. 

Cellular aggregates were identified on days 1, 4, 7 and 14 in 86.7%, 39.5%, 20.0% and 4.4% of eyes, respectively. The positive predictive value of cellular aggregate visibility for successful corneal deturgescence was 71.8%, 64.7%, 66.7% and 100.0% at each time point, respectively (Table 1). AS-OCT images of several representative eyes are presented in Figure 1. A logistic regression model was performed to ascertain the effect of cell visibility on the likelihood of successful corneal deturgescence. The logistic regression model was not statistically significant at any of the time points, χ^2^(1) = 1.08, *p* = 0.299; χ^2^(1) = 0.339, *p* = 0.561; χ^2^(1) = 0.26, *p* = 0.873; χ^2^(1) = 1.53, *p* = 0.216 for days 1, 4, 7 and 14, respectively. We then included CCT as an additional independent variable and re-performed the logistic regression modelling. The logistic regression model reached statistical significance for days 4, 7 and 14, with χ^2^(1) = 5.761, *p* = 0.560; χ^2^(1) = 14.656, *p* = 0.001; χ^2^(1) = 22.702, *p* = 0.000; χ^2^(1) = 22.257, *p* = 0.000 for days 1, 4, 7 and 14, respectively. An increase in pachymetry resulted in a small but statistically significant decreased likelihood of success by an odds ratio of 0.996 (95% CI 0.993–1.000), 0.996 (95% CI 0.993–0.999) and 0.996 (95% CI 0.994–0.998), for days 1, 2 and 14, respectively, and an odds ratio of 0.994 (95% CI 0.991–0.998) for day 7 (Table 2). Based on logistic regression modelling using the cell visibility and pachymetry data, receiver-operating characteristic (ROC) curves were plotted, and the areas under the curve (AUC) calculated for each time point (Figure 2). The AUC values were 0.72 (95% CI 0.55–0.89), 0.80 (95% CI 0.62–0.98), 0.86 (95% CI 0.71–1.00) and 0.90 (95% CI 0.80–0.99) for days 1, 4, 7 and 14, respectively. 

We performed a sensitivity analysis to compare cell visibility and pachymetry between expanded primary CECs (n = 37) and SNECi (n = 8) eyes. We found significant differences in cell visibility on day 4 (*p* = 0.042) and day 7 (*p* = 0.039), with a higher proportion of eyes with cell visibility in the SNECi group (Pearson’s chi-square test). There were no significant differences between the groups in pachymetry at all time points (Mann–Whitney test) (Table 3). There were no differences in the rate of success between the groups, χ^2^(1, N = 45) = 0.185, *p* = 0.667. 

## 4. Discussion

We reported the predictive value of cell visibility and pachymetry on AS-OCT imaging for corneal deturgescence in a rabbit model of bullous keratopathy. The visibility of cellular aggregates alone did not appear to have a statistically significant effect on the prediction of corneal deturgescence. Including the CCT data in the logistic regression model, a decrease in pachymetry on days 4, 7 and 14 did significantly predict the eventual corneal deturgescence. ROC curves plotted from the logistic regression model that included both cell visibility and CCT data demonstrated AUC with fair (day 1) and good (day 4, 7 and 14) prediction. 

Huang et al. first reported the use of a low-coherence imaging device that enabled in vivo, non-contact imaging of ocular structures in 1991 [37]. OCT imaging of the cornea was described soon after by Izatt et al [38]. As the technology has experienced refinements, including advances in image quality, axial resolution and speed of acquisition, the range of applications in diagnosis and management of anterior segment pathology has also grown. AS-OCT has been demonstrated to have utility in differentiating ocular surface lesions [39], aiding the clinical assessment of infectious keratitis and thinning [40], keratoconus [41] and corneal dystrophies, and has become an essential tool in ophthalmology departments [42]. AS-OCT also provides critical information when evaluating surgical corneal patients, particularly with the established trend toward selective lamellar surgery [43]. In endothelial keratoplasty, particularly DMEK, it can be challenging to determine graft attachment by slitlamp examination alone. AS-OCT is an invaluable aid in these situations and may reduce the rates of primary or immediate post-operative graft failure. Cell therapy for corneal endothelial disease is an emerging area of research that has the potential to revolutionize the field [44]. In this study, we reported for the first time the novel application of AS-OCT to attempt to predict a successful cell injection in a model of bullous keratopathy. 

Following CECI, the rabbits remained anaesthetized and were postured for three hours such that the treated eye was in a prone position. This was to maximize the probability that the injected cells settled inferiorly and adhered to the host DM. Cellular aggregates were detectable on AS-OCT imaging in a significant percentage of eyes (86.7%) on day 1, and this percentage decreased over the course of the follow-up. It was hypothesized that the visibility of cellular aggregates could be an indicator of cellular adherence to the host DM and, therefore, might be an early predictor of success. In vivo CECs are approximately 5 μm thick and 20 μm in diameter [45]. As the RTVue has an axial resolution of 5–7 μm [46], there is an inherent limitation in the ability of the AS-OCT to capture the presence of donor cells lining the host DM. Logically speaking, the absence of cellular aggregates on imaging could be due to: 1. inadequate sampling—only a single horizontal B-scan aimed to visualize the central cornea was acquired; 2. a diffuse distribution pattern along the DM as the cells might not be visible without aggregation due to their size; 3. a paucity of adherent cells; 4. a complete non-adherence or absence of cells; 5. a combination of the previous reasons. For example, animal J97 did not have any detectable cellular aggregates on day 1 and yet demonstrated good deturgescence by day 7. This indicated the adherence and functionality of donor CECs despite not being ‘seen’. Furthermore, the percentage of eyes in which cellular aggregates were visible decreased dramatically over the observation period (Figure 3). This corroborates previous reports suggesting that multilayer CEC deposits detached over time and were washed out through the trabecular meshwork [47]. 

The converse was also observed: in 11 eyes, cellular aggregates were visualized at early time points but this failed to result in corneal clarity. These eyes may represent non-functional adhered CECs. Bostan et al. reported the phenomenon of the formation of a multilayer endothelium with myofibroblastic transformation in a feline model of bullous keratopathy treated with cell injection [47]. These eyes resulted only in partial corneal deturgescence. The injected CECs may have adhered early but subsequently detached. The rabbits received a depot injection of subconjunctival dexamethasone at the end of the procedure, followed by topical antibiotic and steroid eyedrops four times a day for the duration of the follow-up. However, we are unable to rule out the possibility of subclinical rejection due to inadequate immunosuppression. Finally, these eyes may represent a limitation of this particular rabbit model of bullous keratopathy, in which the DM is debrided limbus to limbus, and the injected CECs are required to form a monolayer covering the entire bare DM. In a human patient, this would be limited to the central 7–8 mm. 

The current surgical approach of endothelial keratoplasty to treat endothelial failure is unlikely to ever be adequate to treat the many millions of individuals worldwide in need of this procedure [6]. Alternative solutions that overcome the constraints of donor shortage are required, and numerous options are being investigated. These include regenerative medicine approaches, such as Descemet Stripping Only (DSO) [48], synthetic approaches such as the EndoArt (EyeYon Medical, Ness Ziona, Israel), an artificial endothelial layer [49], and a plethora of bioengineering approaches that have been comprehensively reviewed elsewhere [44,50,51,52]. CECI is one such promising solution in a growing field, but questions remain regarding its long-term efficacy and safety. For example, the 11 patients in the seminal clinical study of CECI by Kinoshita et al. received either 5 × 10^5^ or 1 × 10^6^ cells, and there are legitimate concerns about the theoretical risk of both senescence-related inflammation and tumorigenicity of cells that do not adhere to the DM [16,53]. While CECI is a leading option for corneal endothelial-based therapy, these other approaches may remove the primary long-term concern with cell injection therapy. Our study adds to the current understanding of CECI by reporting the novel use of an in vivo imaging technique to assess for the presence of cellular aggregates adhering to the posterior stroma following CECI. However, further study is warranted to determine the fate of the injected CEC, and this is beyond the scope of this discussion. In human subjects, a decade-long follow-up will ultimately be needed.

Our culture methodology differs from that described in the paper by Kinoshita et al [29] in a number of ways. Briefly, our culture method utilized a dual media culture system in which CEC were first established in an endothelial-based stabilization medium containing 5% serum. Subsequently, the medium was switched to a proliferative medium formulated using M199 and Ham’s F12 mixed in a 1:1 ratio, 5% serum, basic fibroblast growth factor and a rho-associated kinase inhibitor (Y-27632). In contrast, Kinoshita et al. used a single-medium culture system with Opti-MEM, 8% fetal bovine serum, epidermal growth factor, Y-27632, p38 MAP kinase inhibitor (SB 203580), as well as a transforming growth factor beta inhibitor (SB 431542) for their expanded cultures. In summary, Kinoshita et al. utilized a highly proliferative single-medium culture system, compared to our dual media highly–lowly proliferative culture system. However, in both studies, the expanded CEC were dissociated into single cells, injected into the anterior chamber and left undisturbed for 3 h to adhere to the Descemet membrane. The differences in culture methodology should not affect our study results, which were obtained using AS-OCT. 

The comparison between expanded primary CEC and SNECi cell preparation methods showed significant differences in cell visibility on days 4 and 7 in favor of SNECi. However, there were no differences in the rates of success or failure. The SNECi approach is a recently described method in which single cells are isolated from cadaveric donors but do not undergo in vitro expansion [26]. As the SNECi cell would be a Class 1 biological product, this simplified technique has the potential to bypass stringent regulatory requirements and the use of Good Manufacturing Practice-accredited facilities, due to the minimal manipulation of cells. Future studies investigating differences in efficacy between the two techniques of CEC preparation are warranted.

Our pilot study has a number of limitations. Firstly, the small sample size may have resulted in a type II error, failing to reach statistical significance. However, this is still a relatively new treatment modality. Secondly, image acquisition was performed using an isolated horizontal B-scan. Performing a raster scan will provide a more comprehensive assessment and likely increased detection of cellular aggregates across a larger area of the corneal endothelial surface. Further studies with larger cohorts and increased sampling of the corneal endothelium with AS-OCT imaging are needed. Histological analysis would also be of benefit to determine the underlying reasons behind unsuccessful corneal deturgescence. Finally, we used a rabbit model of bullous keratopathy which has inherent limitations. In particular, the rabbit endothelium has a much higher regenerative capacity compared to the human endothelium [54]. To address this, endothelial cells were debrided limbus-to-limbus, and the Descemet membrane stained with vision blue to assist in the complete removal of the host endothelium. Additionally, the rabbits were sacrificed after day 14 post-CECI, as it could be expected that any corneal clearing at later timepoints could be due to host endothelial regeneration. Mouse models have been proposed as an alternative model for cell injection. Apart from cell injection being more challenging to perform in rodent eyes, there remains the fact that, involving a xenograft, this procedure would require long-term immunosuppression, as episodes of rejection would negatively impact the final endothelial cell counts. Non-human primate models exhibit a similarly poor regenerative capacity to human corneas and are more appropriate to study the long-term efficacy of the treatment, and we have an ongoing study using this model.

## 5. Conclusions

In conclusion, we investigated the predictive value of cell visibility and pachymetry on AS-OCT imaging for corneal deturgescence in a rabbit model of bullous keratopathy. While the visibility of cellular aggregates alone did not appear to have a statistically significant effect on the prediction, logistic regression modelling including the CCT data was predictive of the eventual corneal deturgescence. 

## Figures and Tables

**Figure 1 cells-12-01167-f001:**
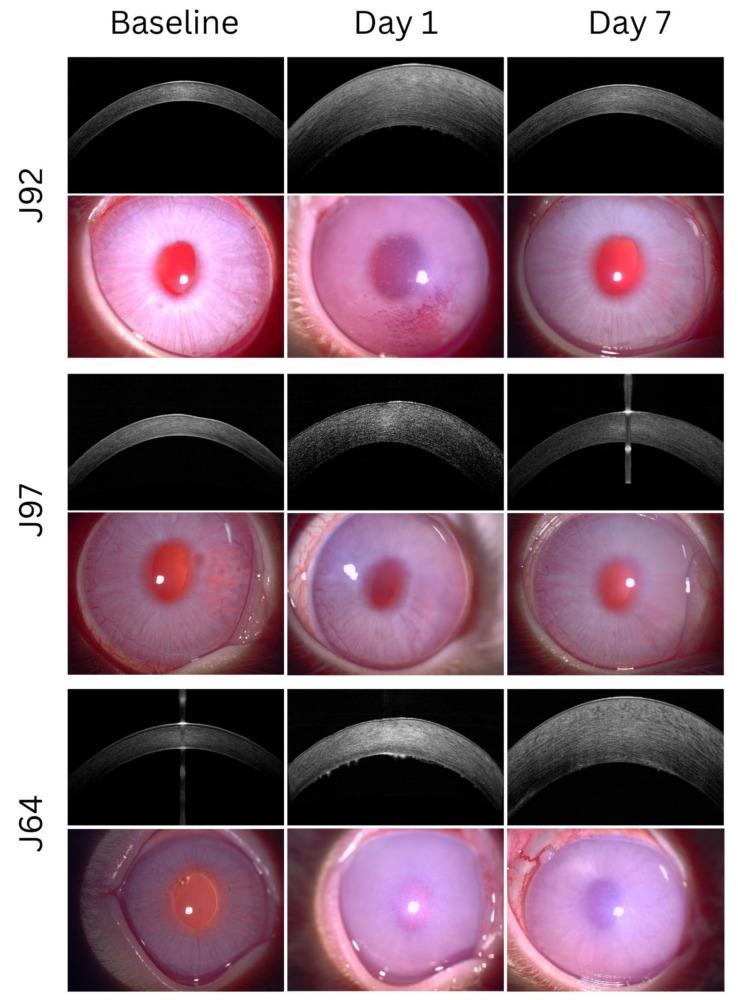
AS-OCT and slitlamp images of three animal eyes. J92 is a representative image set of successful deturgescence with cell visibility on day 1 and corneal clearing by day 7. J97 is a representative image set of successful deturgescence with no cell visibility on day 1, and corneal clearing by day 7. J64 is a representative image set of unsuccessful deturgescence with cell visibility on day 1.

**Figure 2 cells-12-01167-f002:**
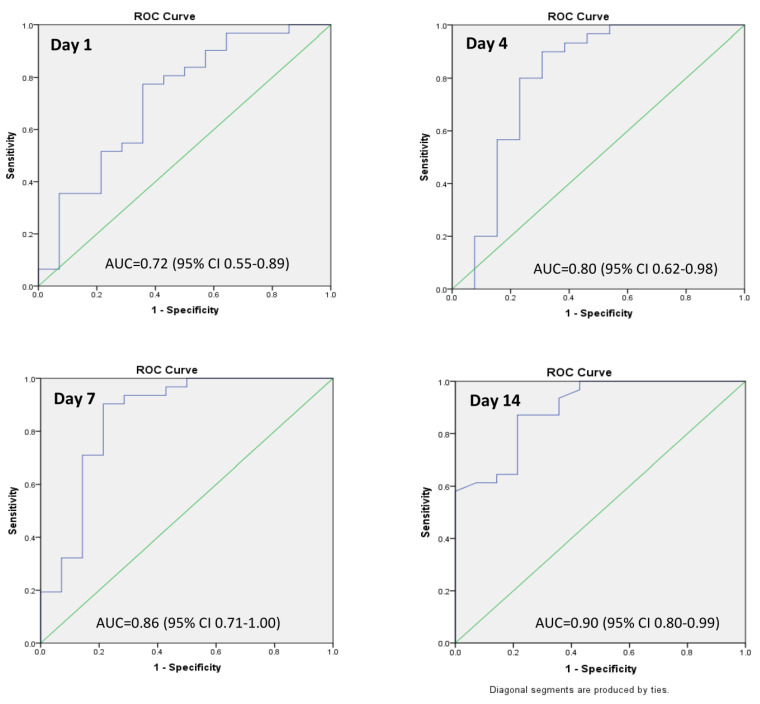
Receiver operating curves (ROC) based on logistic regression modelling using cell visibility and pachymetry data for days 1, 4, 7 and 14.

**Figure 3 cells-12-01167-f003:**
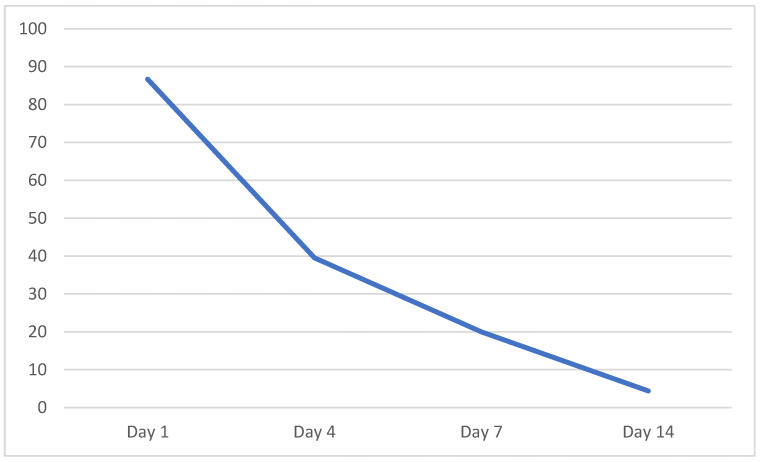
Percentage of eyes with visibility of cellular aggregates on AS-OCT imaging over time.

**Table 1 cells-12-01167-t001:** The predictive value of cell visibility for successful corneal deturgescence.

	Day 1	Day 4	Day 7	Day 14
Sensitivity	0.903	0.367	0.194	0.065
Specificity	0.214	0.538	0.786	1.000
Positive predictive value	0.718	0.647	0.667	1.000
Negative predictive value	0.500	0.269	0.306	0.326
Positive likelihood ratio	1.150	0.794	0.903	NA
Negative likelihood ratio	0.452	1.176	1.026	0.935

**Table 2 cells-12-01167-t002:** Logistic regression modelling of the effect of cell visibility and pachymetry on successful corneal deturgescence and odds ratio of increased CCT predicting a decreased likelihood of successful corneal deturgescence.

Cell Visibility	Chi-Square	df	*p*-Value
Day 1	1.080	1	0.299
Day 4	0.339	1	0.561
Day 7	0.260	1	0.873
Day 14	1.532	1	0.216
**Cell visibility and CCT**			
Day 1	5.761	1	0.560
Day 4	14.656	1	0.001
Day 7	22.702	1	0.000
Day 14	22.257	1	0.000
**CCT**	**Odds ratio**	**95% CI**	
Day 1	0.996	0.993–1.000	
Day 4	0.996	0.993–0.999	
Day 7	0.998	0.991–0.998	
Day 14	0.996	0.994–0.998	

CCT—central corneal thickness (μm); df—degrees of freedom; CI—confidence interval.

**Table 3 cells-12-01167-t003:** Sensitivity analysis comparing eyes receiving primary expanded CECs or SNECi CECs.

		Cell Visibility			
		SNECi CECs	Primary Expanded CECs		Exact Sig. (2-Sided) *p*-Value
Day 1	N	0 (0%)	6 (16.2%)		0.572
	Y	8 (100%)	31 (83.8%)		
Day 4	N	2 (25%)	24 (68.6%)		0.042
	Y	6 (75%)	11 (31.4%)		
Day 7	N	4 (50%)	32 (86.5%)		0.039
	Y	4 (50%)	5 (13.5%)		
Day 14	N	8 (100%)	35 (94.6%)		1.000
	Y	0 (0%)	2 (5.4%)		
Pachymetry					
	**SNECi CECs**		**Primary Expanded CECs**		
	**N**	**Median (IQR)**	**N**	**Median (IQR)**	**Asymp. Sig. (2-Tailed) *p*-Value**
Baseline	8	351 (322–375)	34	374 (352–395)	0.065
Day 1	8	842 (625–1012)	37	720 (641–866)	0.467
Day 4	8	884 (726–1360)	37	652 (460–987)	0.085
Day 7	8	785 (538–1020)	37	517 (461–873)	0.109
Day 14	8	571 (486–1525)	37	635 (489–1225)	0.941

## Data Availability

The data presented in this study are available on request from the corresponding author.

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
