# Peer review of "Early Visibility of Cellular Aggregates and Changes in Central Corneal Thickness as Predictors of Successful Corneal Endothelial Cell Injection Therapy"

_cells, 2023, doi:10.3390/cells12081167_

Round 1

Reviewer 1 Report

The authors investigated the predictive value of cell visibility and pachymetry on AS-OCT imaging on corneal deturgescence in a rabbit model of bullous keratopathy. CECI is a very good solution to make up for the global shortage of transplant-grade corneal tissue. The paper is commendable for Logistic regression models of cell aggregation visibility and CCT to predict the success of corneal endothelial cell injection therapy, but there are still some detailed problems to be rectified and revised:

1.     Writing/Text issues: for example, there are two “.” at the end of “Abstract” in line 34 and please check the full text carefully.

2.     Are all pictures have the same magnification (Figure 1)?

3.     Can authors provide photographs of the anterior segment of rabbits in the corresponding days in Figure 1 ?

4.     Rabbit CECs have a certain proliferation capacity, the CECs should be labeled prior to injection, and histological observation before injection and 14 days after injection should be provided.

Reviewer 2 Report

See uploaded document.
